# Amino Acid Transporters in Plants: Identification and Function

**DOI:** 10.3390/plants9080972

**Published:** 2020-07-31

**Authors:** Xuehui Yao, Jing Nie, Ruoxue Bai, Xiaolei Sui

**Affiliations:** Beijing Key Laboratory of Growth and Developmental Regulation for Protected Vegetable Crops, College of Horticulture, China Agricultural University, Beijing 100193, China; xuehuiyao@cau.edu.cn (X.Y.); jnie@cau.edu.cn (J.N.); bairx@cau.edu.cn (R.B.)

**Keywords:** amino acid transporter, export, phloem loading and unloading, plant, root uptake, UMAMIT

## Abstract

Amino acid transporters are the main mediators of nitrogen distribution throughout the plant body, and are essential for sustaining growth and development. In this review, we summarize the current state of knowledge on the identity and biological functions of amino acid transporters in plants, and discuss the regulation of amino acid transporters in response to environmental stimuli. We focus on transporter function in amino acid assimilation and phloem loading and unloading, as well as on the molecular identity of amino acid exporters. Moreover, we discuss the effects of amino acid transport on carbon assimilation, as well as their cross-regulation, which is at the heart of sustainable agricultural production.

## 1. Introduction

Nitrogen (N) is an essential nutrient for plant growth and reproduction. Plants take up both inorganic nitrogen (ammonium and nitrate) and organic nitrogen (amino acids, peptides, proteins, and other *N*-containing compounds) from the soil [1,2,3,4]. Following their uptake, *N* assimilation comprises the reduction of nitrate to ammonium and the *N* in ammonium is reduced to amino acids in the root. Alternatively, nitrate for reduction to amino acids in the photosynthetically active source leaves after their translocation via the xylem [5,6]. The first organic nitrogenous molecule produced from inorganic *N* is glutamine or glutamic acid, which is then transaminated to produce other amino acids or *N*-containing compounds [7]. Most proteinogenic amino acids are synthesized in the plastids of mesophyll cells, but they can also be manufactured in other cellular compartments, such as mitochondria, peroxisomes, and the cytosol [6]. In most species, asparagine, and glutamine are most abundant in the xylem sap, whereas all amino acids are transported through the phloem [8,9,10]. However, within the xylem and phloem transport systems, the concentration of individual amino acids may vary, depending on plant species and environmental conditions. The resulting pool of amino acids represents the main form of organic *N* exported to sink organs, such as root tips, flowers, growing leaves, fruits, and seeds, to sustain growth and development [6,11]. Therefore, amino acid transporters are essential when moving amino acids in or out of plant cells, as well as various compartments (i.e., chloroplast, peroxisome, mitochondrion, and vacuole) for the proper distribution of organic *N* throughout the plant [12,13].

Amino acid transporters fall into two families in plants based on their sequence similarity and uptake properties: The amino acid/auxin permease (AAAP) family, also called the amino acid transporter (ATF) family, and the Amino Acid-Polyamine-Organocation (APC) family [14]. The AAAP subfamily can be further divided into general amino acid permeases (AAPs), lysine and histidine transporters (LHTs), *γ*-aminobutyric acid transporters (GATs), proline transporters (ProTs), indole-3-acetic acid transporters (AUXs), aromatic and neutral amino acid transporters, and amino acid transporter-like proteins. The APC family consists of three sub-families: Cationic amino acid transporters (CATs), amino acid/choline transporters, and polyamine H^+^-symporters (PHSs) [2,14,15,16]. Another group of transporters, designated ‘usually multiple acids move in and out transporters’ (UMAMIT), was recently identified in Arabidopsis (*Arabidopsis thaliana*) [17,18] (Appendix A).

Amino acid transporters have been identified and analyzed in several model or crop species, including Arabidopsis, tomato (*Solanum lycopersicum*), potato (*Solanum tuberosum*), broad bean (*Vicia faba* L.), barley (*Hordeum vulgare* L.), maize (*Zea mays* L.), pea (*Pisum sativum* L.), rice (*Oryza sativa* L.), and common bean (*Phaseolus vulgaris* L.) [19,20,21,22,23,24,25,26,27]. The first plant amino acid transporter to be reported, Amino Acid Permease 1 (AAP1), was identified on the basis of functional complementation of a yeast amino acid transporter mutant by heterologous expression of an Arabidopsis cDNA library [28,29]. With the development of genomics resources, genome-wide surveys have identified putative amino acid transporters in Arabidopsis (at least 60 genes), wheat (*Triticum aestivum*, 85 genes), rice (189 genes), soybean (*Glycine max*, 72 genes), potato (100 genes), poplar (*Populus trichocarpa*, 23 genes), *Selaginalla* (62 genes), and castor bean (*Ricinus communis*, 283 genes) [13,15,26,30,31,32,33,34]. That these transporters exhibited biochemical properties comparable to Arabidopsis homologues suggested that the function of amino acid transporters is conserved across vascular plants. However, the role of each amino acid transporter in organic *N* partitioning is far from understood. Most of the transporters characterized, thus far, localize to the plasma membrane and function as proton-coupled importers or exporters between cells (namely intercellular transport) [13]. By contrast, only a few transporters mediating intracellular (namely vacuole, chloroplast, and mitochondrion) transport of amino acids have been described [35,36,37,38,39,40,41]. Transporters generally differ in substrate selectivity and affinity when analyzed in heterologous expression systems (budding yeast *Saccharomyces cerevisiae*, or frog (*Xenopus laevis*) oocytes). In addition, homologous transporter genes exhibit distinct tissue-specific expression profiles between various plant species [19]. As interest in the field of amino acid transport grows, the physiological and genetic functions of amino acid transporter genes commonly rely on the characterization of mutants and transgenic overexpression lines.

The uptake and distribution of organic *N* in plants has been studied for many years and has been covered by several recent and older reviews [1,2,5,6,7,12,13,14,16,42,43,44]. In this review, we discuss recent advances describing the identification, function, and regulation of amino acid transporters in plants, as well as possible directions for future research.

## 2. Amino Acid Uptake, Transport, and Distribution

### 2.1. Transporters Mediating Root Amino Acid Uptake

Low- and high-affinity transporters mediate amino acid uptake from the soil. Following uptake or biosynthesis in the roots, amino acids then move from root hairs or epidermal cells to the vascular cylinder via the symplasm. Alternatively, transporter-mediated import into the root symplasm takes place at or before the endodermis, since the Casparian strip blocks apoplastic flow to the root vasculature. For xylem loading, amino acids are released into the apoplasm from the endodermis, pericycle, or xylem parenchyma cells using export proteins.

In Arabidopsis, at least five amino acid transporters play a role in amino acid uptake in roots and belong to three families within the AAAP group: AAPs, LHTs, and ProTs. Arabidopsis AAPs and LHTs are broad substrate transporters for neutral and acidic amino acids, whereas ProTs specifically transport proline, glycine, and γ-aminobutyric acid (GABA) [45,46,47]. Arabidopsis AAP1 localizes to the root tip and epidermal cells, including root hairs, and transports glutamate and neutral amino acids [45,48,49]. Arabidopsis AAP5 is expressed in the root and functions in the acquisition of basic amino acids [50,51,52], whereas Arabidopsis AAP3, expressed in the root vascular tissue, may be involved in amino acid uptake from the phloem or the soil [53]. LHTs are considered to be high-affinity transport systems. Chen and Bush [54] documented expression of Arabidopsis LHT1 at the root surface and assigned it a lysine- and histidine-selective transporter function, although other studies described a role in the uptake of neutral and acidic amino acids into roots for LHT1 [55,56,57]. In addition, histochemical analysis of p*LHT1:GUS* (*ß-GLUCURONIDASE*) reporter lines revealed that Arabidopsis *LHT1* is preferentially expressed in the lateral root cap [55]. The Arabidopsis root expression map detects *LHT1* expression in the root epidermis, cortex, and endodermis during early plant development, which supports a direct role for LHT1 in amino acid import into root cells [58]. In rice, *LHT1* is expressed throughout the root, including root hairs, the epidermis, cortex, and stele, as demonstrated by GUS reporter lines. Knockout of *OsLHT1* by genome editing in japonica rice exhibited reduced root uptake of amino acids [59]. Other studies showed that rice *AAP3* and *AAP6* are expressed in the elongation zone of lateral roots, root stele, and epidermis, and function as regulators of amino acid levels in roots [60,61]. Arabidopsis LHT6 is highly expressed in root cells and contributes to the assimilation of acidic amino acids, glutamine, and alanine from the rhizosphere [49]. In addition, Arabidopsis *ProT2* is expressed in the root epidermis and cortex, where the encoded protein functions in the import of the compatible solutes proline and glycine betaine [47,62,63,64,65].

Based on the specific expression patterns mentioned in the above studies of *AAP* and *LHT* genes in the root, we generated a model of amino acid uptake in the root (Figure 1). However, the expression profile of individual transporters does not explain all aspects of amino acid uptake, as several studies have confirmed that transporter activity in the root may vary depending on soil conditions and plant species. For example, LHT1 and AAP5 are crucial for amino acid uptake at concentrations in the soil below 50 μM. Transport studies with *aap1* mutants suggest that AAP1 may take up amino acids at high concentrations [48,51,66]. In addition, AAP1 functions in the acquisition of glutamate and neutral amino acids when present in the soil at ecologically-relevant concentrations, whereas LHT6 is involved in the import of the acidic amino acids alanine and aspartate by roots at both low and high concentrations [49].

### 2.2. Transporters Function in Xylem–Phloem Transfer and Intercellular Transport of Amino Acids

The identification of amino acid transporters is essential to understand how they regulate *N* root uptake, as well as root-to-shoot and leaf-to-sink transport. Root-to-shoot movement of amino acids occurs in the xylem, whereas amino acid partitioning from source leaves to sink organs takes place in the phloem. However, some amino acids can also be removed from the long-distance transport mediated by the xylem and transferred to the phloem to supply fast-growing sink organs, such as root tips and young leaves [67,68]. This transfer between xylem and phloem requires the retrieval of amino acids from the transpiration stream (xylem) to xylem parenchyma cells, with subsequent symplastic movement to phloem parenchyma cells. Ultimately, amino acids are released into the phloem sap [5,44,69].

Arabidopsis AAP6 localizes to the xylem parenchyma, where it mediates *N* exchange between the xylem and phloem, as evidenced by the reduced amino acid concentrations in the phloem of *aap6* mutants [70]. In addition, Arabidopsis AAP2 is expressed in phloem companion cells along the transport path, and *aap2* mutants displayed reduced organic *N* supply to seed sinks, leading to reduced seed protein levels [71] (Figure 1). Once in the leaf, amino acids are imported into parenchyma or mesophyll cells surrounding the xylem by the action of Arabidopsis LHT1 [55].

In addition to long-range transport, amino acids synthesized inside cells move across various organelles, which requires intracellular transporters. Fusion proteins between transporters and the green fluorescent protein (GFP) have demonstrated that several amino acid transporters localize to organellar membranes rather than the plasma membrane (Figure 1). The identification of vacuole transporters is the focus of much research in multiple plant species. In Arabidopsis, the cationic amino acid transporters CAT2 and CAT4 localize to the vacuolar membrane (tonoplast), and CAT2 is implicated in the regulation of amino acid levels in leaves [72,73]. Arabidopsis CAT8 localizes to both the plasma membrane and the tonoplast [74]. In tomato, CAT9 was identified using quantitative proteomics of a tonoplast-enriched membrane fraction. Tomato CAT9 is a tonoplast exchanger that transports glutamine and aspartate into the vacuole lumen in exchange for *γ*-aminobutyrate (GABA), and plays a role in amino acid accumulation during fruit development [75]. Another tomato member of the CAT family, CAT2, localizes to the tonoplast in stamen cells, indicating a role in flower development [76]. Outside of the CAT family, members of the Amino acid Vacuolar Transport (AVT) sub-group may also function at the tonoplast, since homologues from yeast can mediate amino acid transport across the vacuolar membrane [77]. One of the Arabidopsis homologues, AVT3, transports alanine and proline from the vacuole into the cytosol when expressed in yeast [78]. In addition, the Arabidopsis transporters DICARBOXYLATE TRANSPORT (DiT2.1) and the MITOCHONDRIAL BASIC AMINO ACID CARRIERs (mBAC) mBAC1 and mBAC2 localize to the chloroplast and mitochondrial membranes. DiT2.1 functions in malate/glutamate exchange during photorespiration [37]. The two mBACs transport arginine, lysine, ornithine, and histidine by an exchange mechanism [38].

### 2.3. Transporter Function in Phloem Loading of Amino Acids

In leaves, amino acids are synthesized from inorganic *N* and photosynthates. Alternatively, amino acids are also synthesized from photorespiration and the hydrolysis of leaf proteins [2]. Following synthesis, amino acids are released into the cytosol by transporters, transported in the phloem to sink tissues, or stored in the vacuole [13]. To be exported out of leaves, amino acids are loaded into the SE/CC complexes of minor veins. Loading of amino acids into the sieve elements and companion cells of the phloem may follow an apoplastic or symplastic route, depending on the plant species and the number of functional plasmodesmata connecting phloem parenchyma and companion cells [44,79]. In the symplastic pathway, amino acids diffuse between cells through plasmodesmata towards the phloem. During exoplasmic loading, amino acids first need to be released into the cell wall space and subsequently taken up by neighboring cells. This pathway relies on plasma membrane-localized amino acid transporters [43,44,80]. The AAP family of transporters has been proposed to facilitate import into the phloem [81], whereas the bidirectional transporters SIAR1/UMAMIT18 (siliques are RED1/ usually multiple acids move in and out transporters) and bidirectional amino acid transporter 1 (BAT1) may mediate amino acid export from leaf cells [17,39].

It is currently believed that AAPs play a major role in phloem loading, as they transport a broad spectrum of amino acids, although their exact function remains to be investigated (Figure 1). In Arabidopsis, *AAP1*, *AAP2*, *AAP3*, *AAP4*, *AAP5*, and *AAP8* are expressed in mature leaves and may be involved in the phloem loading process [2,19,82,83,84]. *AAP1* and *AAP4* are present in the phloem of leaf minor and major veins, whereas *AAP8* is expressed in source leaves during the vegetative and reproductive phases. Moreover, the *aap8* mutant reduces source-to-sink transport of amino acids, demonstrating that AAP8 is indeed fundamental for the loading of a broad spectrum of amino acids into the phloem to supply sink organs with essential *N* [84,85].

A potential role of AAPs in phloem loading has also been observed in other plants. In pea, overexpression of AAP1 increased phloem loading of amino acids, resulting in improved source-to-sink *N* transport, enhanced sink organ development, and higher seed yield [86]. Based on expression and localization studies, members of other transporter families are also suspected to function in phloem loading (Figure 1). This includes the Arabidopsis transporters CAT1, CAT6, and CAT9, as well as members of the ProT family [12,47,63,65,73,87]. By contrast, AROMATIC AND NEUTRAL TRANSPORTER1 (ANT1) may participate, directly or indirectly, in phloem loading, as the amino acid content of ant1 mutant sieve tubes rose sharply over wild-type levels [88].

### 2.4. Transporter Function in Phloem Unloading of Amino Acids in Sink Organs

Transport of amino acids between phloem and surrounding tissues either follows apoplastic phloem mechanism or the symplastic pathway. Organic *N* needs to be released from one cell and subsequently been taken up by the neighboring cells in apoplastic phloem unloading. Symplastic transport is likely to be rate-limited by movement through plasmodesmata. [5,80,89,90,91]. However, transport will follow either a symplastic or apoplastic route depending on the plant species, sink tissues, and developmental stage. For example, transporters might move *N* across membranes of different cell layers in terminal sinks, such as flowers, fruits, or seeds [80,89,92,93]. Based on expression and localization studies, several amino acid transporters localize to floral structures, suggesting an essential role in N supply for male and female gametophyte development [94,95,96]. In Arabidopsis, CAT, AAP, BAT, and ProT-type transporters import amino acids into flower tissues. Arabidopsis LHT2 and LHT4 are expressed in the tapetum, suggesting their role in delivering amino acids to pollen grains [95,97]. In addition, Arabidopsis *LHT5* and *LHT6* expression was detected along the transmitting tract of the pistil and the pollen tube, pointing to a function in amino acid uptake for successful fertilization [95]. In tomato, high *ProT1* expression was detected in mature and germinating pollen, suggesting that LeProT1 might be involved in pollen nutrition [20].

In seeds, phloem unloading to the seed coat occurs symplasmically [44,93]. In addition, fluorescent tracers demonstrated symplastic continuity between terminal sieve elements and their neighboring cells, which define a region called the unloading domain [98,99]. Despite the symplastic continuum, amino acid transporters nevertheless accumulate in this zone. Amino acid release from the phloem occurs in two different pathways—the symplastic pathway and the apoplastic pathway. Transporters of the UMAMIT type contribute to bidirectional amino acid transport and accumulation in developing seeds. Indeed, Arabidopsis UMAMIT11, 14, and 18 are involved in amino acid export from the phloem to developing seeds [17,100,101] (Figure 1). An analysis of *umamiT11* and *umamiT14* mutants provided support for the physiological relevance of phloem unloading in amino acid export in the developing seed, as these mutants accumulated free amino acids in fruits and produced smaller seeds [100]. Moreover, studies have revealed the essential roles of Arabidopsis AAP1, AAP8, CAT5, and CAT6 during amino acid allocation to the embryo to sustain seed development and the accumulation of storage proteins. Specifically, Arabidopsis AAP1 mediates amino acid uptake to the embryo: *aap1* mutant embryos display reduced amino acid import, resulting in lower protein content [102]. Although *aap1* loss-of-function plants affect seed *N* and protein content, the amino acid transporter CAT6, expressed in embryos, can (to some extent) compensate for the loss of AAP1 [87]. Promoter–GUS analysis revealed that *CAT5* transcription specifically occurs in seeds, suggesting that CAT5 may contribute to seed development [73]. Similarly, Arabidopsis AAP8 plays an essential role in the import of amino acids to the endosperm [103]. In broad bean, the genes encoding the transporters AAP1 and AAP3 are expressed in cotyledons and in the seed coat, and might play an important role in providing amino acids during seed development. AAP1 plays an important role in providing amino acids for storage protein biosynthesis, while AAP3 may have a role in seed coat unloading. Consistent with this hypothesis, overexpression of broad bean AAP1 in pea seeds resulted in increased accumulation of storage proteins [22].

## 3. The Role of Exporters in Amino Acid Translocation

The past three decades have witnessed the discovery of multiple amino acid importers. In addition, physiological studies have revealed amino acid export activity in plant cells, although little is known about the molecular identity of the relevant exporters. The identification of amino acid exporters is essential to complete the circle of amino acid cycling in plants. Specific root-specific exporters release amino acids into the rhizosphere or to root nodules, although little is currently known about the identity of the transporters involved in root amino acid exudation. However, a number of physiological studies have suggested that amino acid efflux might not rely on dedicated transporters, but may instead simply represent leakage from the root [1,104].

Amino acid efflux systems are required at several stages of plant development: Xylem loading, amino acid release into the leaf apoplastic space, and unloading of sink organs. The discovery of UMAMIT transporters has opened the door to a better understanding of amino acid export between cells and across organelles. In Table 1, we provide an overview of the genes encoding amino acid exporters. The first described exporter was SIAR1/UMAMIT18, involved in cellular efflux of glutamine and histidine. A follow-up study showed that UMAMIT18 is also implicated in exporting amino acid to developing siliques [17]. As mentioned earlier, other members of the UMAMIT transporter family are expressed in siliques and export amino acids to the developing embryo. For example, UMAMIT11 and 14 were detected in the unloading domain throughout seed development, whereas UMAMIT28 only accumulated later during seed development. In addition, UMAMIT29 protein was detected in the middle layer of the three-layered inner integument, from early stages in embryo development until the late torpedo stage [100]. UMAMIT24 and UMAMIT25 also function in amino acid transfer in developing seeds. UMAMIT24 may play a role in the temporary storage of amino acids, whereas UMAMIT25 may mediate amino acid export from the endosperm [101]. In addition, UMAMIT14 and UMAMIT18 were assigned an exporter function: They localize to the root with a potential role in phloem unloading. Loss-of-function *umamit14* and *umamit18* mutants lead to a reduction in shoot-to-root and root-to-rhizosphere transfer of amino acids that originated in leaves [8]. Arabidopsis BAT1 mediates the import of arginine and alanine, and export of lysine and glutamate in yeast, suggesting that BAT1 is a bidirectional amino acid transporter [39]. Free amino acids accumulate in the sieve tubes of the *ant1* mutant, supporting a function for ANT1 in moving amino acids out of the phloem [88]. Finally, the GLUTAMINE DUMPER1 (GDU1) and LOSS OF GDU2 (LOG2) proteins may regulate amino acid efflux by activating non-selective amino acid carriers. In plants, LOG2 physically interacts with and ubiquitylates the GDU1, suggesting that GDU1 appears to be a necessary activator (characterized by a membrane domain and the family signature amino acid motif Val-Ile-Met-Ala-Gly) involved in the amino acid export. In addition, the overexpression of GDU1 plants displays a large increase of free amino acids throughout the plant, suggesting that GDU1 may be involved in the regulation of amino acid transport [105,106,107,108].

## 4. Is the Functional Overlap of Multiple Amino Acid Transporters Redundant or Necessary?

Members of large multi-gene families often display overlapping functions. Since many amino acid transporters act on the same substrates, specificity may derive from their expression patterns and their responses to environmental signals [14]. For example, the Arabidopsis AAP1 and AAP2 transporters display a preference for neutral and acidic amino acids. However, an analysis of promoter–GUS reporter lines revealed that *AAP1* was expressed in the developing endosperm and cotyledons, thus strongly suggesting that AAP1 functions in the import of amino acids into the endosperm embryo. By contrast, *AAP2* was highly expressed in the vascular tissue of stems and siliques, suggesting a role in amino acid retrieval and uptake into seeds. Moreover, although the Arabidopsis ProT1 and ProT2 proteins both transport the compatible solute proline, the expression pattern of their encoding genes responds differently to water and salt stress [63].

Nonetheless, not all related transporters share the same substrate specificity, as suggested by the characterization of selected amino acid transporters that differ both in their range of substrates and in their site(s) of action. Recently, Choi showed that Arabidopsis LHT2 can transport 1-aminocyclopropane-1-carboxylate (ACC), a biosynthetic precursor of ethylene, when expressed in *Xenopus* oocytes [112]. In addition, UMAMIT5, also called WALLS ARE THIN1 (WAT1), has been ascribed the function of indole-3-acetic acid exporter from the vacuole to the cytosol, unlike other UMAMIT members [18,109]. Seven members of the UMAMIT family have been suggested to play a role in the seed loading process, but their individual expression patterns differ during this process, suggesting that they play distinct roles in amino acid translocation from maternal to filial tissues. Indeed, the phenotypes associated with single *umamit* loss-of-function mutants are relatively benign, consistent with the involvement of multiple exporters in amino acid transport to developing seeds. Similarly, Arabidopsis LHT6 and AAP1 are expressed in root cells and display overlapping functions in root amino acid uptake. However, they also exhibit different substrate preferences. For example, Perchlik showed that LHT6 is involved in the uptake of acidic amino acids, such as glutamine and alanine, and probably phenylalanine, whereas AAP1 transports neutral amino acids and glutamate when amino acids are present at low concentrations [49]. In the feeding experiments, root uptake of the amino acids alanine, glutamine, proline, serine, glutamate, and phenylalanine significantly decreased in *lht6 aap1* double mutant plants. Interestingly, the growth of the *lht6 aap1* double mutant was not affected when grown on Murashige and Skoog medium, possibly because other amino acid transporters expressed in the root can compensate for their loss [49]. Similarly, Arabidopsis LHT1 and AAP5 assimilate neutral and acidic amino acids, respectively, and displayed non-overlapping specificity in their substrates, suggesting that they have separate roles in amino acid uptake [50].

## 5. Regulation of Amino Acid Transporters in Response to Environmental Stimuli

The regulatory aspects of amino acid transport, as well as the underlying sensing and signaling mechanisms, have been discussed in recent years [7]. These studies included the regulation of multiple enzymes and transporters, as enzymes involved in amino acid biosynthesis are regulated by biotic and abiotic stresses. Environmental signals, such as nutrition, light, salt and drought stress, and nematode, or pathogen attack, also influence the expression of amino acids transporters [7,19,63,113]. Much of what we know about the control of amino acid transporter function is limited to transcriptional regulation, with no experimental evidence as of yet to support post-transcriptional regulation.

Transcriptome analysis revealed that 21 of the rice *AAT* genes (out of a total of 85) were differentially expressed under various abiotic stresses [26]. Another study showed that rice *AAT* transcript levels were highly dynamic under *N*-starvation conditions [114]. Other plants have been the subject of an analysis of the environmental factors that affect amino acid transporter expression. In wheat, the expression of *AAT*-related genes responded differently to salt, heat, and drought stresses, which may allow wheat plants to adapt to a wide range of environmental conditions [15]. High nitrate, ammonium, sucrose, glucose, and amino acids induced Arabidopsis *AAP1* expression [14,19]. Microarray experiments indicated that *AAP6* expression was significantly downregulated by glutamine [115]. It is interesting to note that the expression of broad bean *AAP1* is downregulated by combined high glutamine and sucrose [22]. In addition, the expression of Arabidopsis *ANT1*, *LHT1*, and *ProT2* is also induced by different nitrate concentrations in seedlings [19]. Water and salt stress cause the downregulation of Arabidopsis *AAP4* and *AAP6* expression levels, whereas water stress upregulates *ProT2* expression [63]. Liu et al. demonstrated that Arabidopsis *LHT1* acts as a negative modulator of disease resistance, probably by increasing cytosolic glutamine levels and modulating cellular redox status, and that *LHT1* expression is induced by the SA pathway [116]. Similarly, expression of the trefoil (*Lotus japonicus*) homologue of Arabidopsis *LHT1* was induced by mycorrhizal fungal colonization in the roots [117]. A connection between phytohormones and amino acid transporter expression came from ginseng (*Panax ginseng*), where *LHT1* expression was upregulated in response to treatment with abscisic acid, salicylic acid, and methyl jasmonate [118]. As the above experiments describe, amino acid transporter genes, therefore, exhibit complex expression patterns in response to different environmental stimuli, likely integrated by multiple cis-regulatory elements in their promoters. These results open the door for new discoveries into how plants modulate amino acid transport, and thus, growth and development, in response to fluctuating environmental conditions.

## 6. Transport Function in Carbon Metabolism

Nitrogen is an essential and yet limiting factor for crop yield. In general, *N* partitioning between source and sink organs balances *N* uptake and metabolism in source organs, and transport potential from source to sink organs [5]. Moreover, the balance between carbon (C) and *N* is critical to improving *N* use efficiency (NUE): *N* levels can significantly affect C fixation, as photosynthetic proteins, such as Rubisco and PEP carboxylase, constitute a large *N* sink [119]. Alterations in amino acid phloem loading and unloading can regulate *N* and C acquisition and metabolism. Work on Arabidopsis aap8 mutant plants revealed that photosynthesis and N/C assimilation were significantly downregulated during the reproductive stage, consistent with the proposed balance between *N* and C metabolism [84]. In addition, the expression of Arabidopsis AAP1 is regulated by light, C, and nitrate status [14]. Interestingly, the translocation of amino acids between source and sink organs changed in the Arabidopsis *aap2* mutant background, affecting leaf protein/RuBisCo levels, as well as photosynthetic capacity. Moreover, the changes in C metabolism in *aap2* plants also affected seed content in fatty acids, possibly as a result of altered phloem C levels or C:N ratio [71]. However, independently of *N* nutritional status, *aap2* plants translocate significantly more *N* to leaves, thus promoting leaf growth and increasing effective photosynthetic surface area relative to wild-type plants. Furthermore, increased leaf *N* supply positively affected photosynthetic NUE and C assimilation [120]. In turn, increased expression of genes encoding photosynthesis-related enzymes results in a rise in glutamine levels, which may feed back onto the expression of amino acid transporters [121]. Furthermore, many staple crops are C4 plants with higher NUE than C3 plants. Contrasted to C3 photosynthesis, the C4 photosynthetic pathway is more efficient. Therefore, approaches to improve C4 plant NUE have focused on the genetic manipulation of source-to-sink transport of amino acid-N might provide a promising approach for increasing plant photosynthesis efficiency and productivity. In conclusion, it might be possible to tweak the allocation of amino acids to photosynthesis and to developing sinks by overexpressing amino acid transporters in roots or in sink cells, or by invoking an indirect feedback mechanism, with the goal of improving plant productivity and supporting sustainable agriculture. Moreover, understanding the relationship between C and *N* metabolism, as well as metabolic regulatory networks, will help improve crop yield and stress tolerance.

## 7. Future Perspectives

Amino acid transporters contribute to *N* partitioning and distribution between source and sink organs throughout plant development, and yet little is known about how (or whether) they undergo cross-talk. Future efforts should focus on their post-translation regulation by characterizing their associated protein modifications or protein–protein interactions under biotic and abiotic stresses. Generally, amino acid transporters act on the specificity of substrates, while there seems to be very little about sulfur-containing amino acid (cysteine and methionine) transporters. Arabidopsis UMAMIT14 is a broad substrate transporter for amino acids, whereas not specifically transport cysteine and methionine. However, the LHT1 and GAP1 (General Amino acid Permease1) transport methionine in the yeast growth complementation assay [8]. On the other hand, sulfur-containing amino acid might be metabolized to form tripeptides transported by peptides transporters. In terms of the anabolism, cysteine is involved in the formation of the tripeptide glutathione that might transport via specific transporters [122]. Sulfur-containing amino acids from legume crops play a vital role in human nutrition. Therefore, further attention should be paid to the sulfur-containing amino acid and how they might be transported. Although several amino acid exporter systems have been described in Arabidopsis, their counterparts in other species remain unknown. Amino acid exporters should be the subject of a more immediate focus, as very little is known about their regulation. The integration of innovative methods, such as genome editing, high-throughput sequencing, metabolomics, and high-throughput phenotyping, may allow the identification of additional exporters and begin to shed light on the regulation of amino acid export. More generally, amino acid transporters may represent excellent targets for crop improvement, based on their role in long-distance transport of amino acids to sink organs. Finally, even though amino acid transporters modulate *N* and C metabolism, the identification of the underlying signals remains a vast challenge. Further, the emerging tools of genomics and bio-informatics will allow us to identify the C:N cross-talk signal pathways.

## Figures and Tables

**Figure 1 plants-09-00972-f001:**
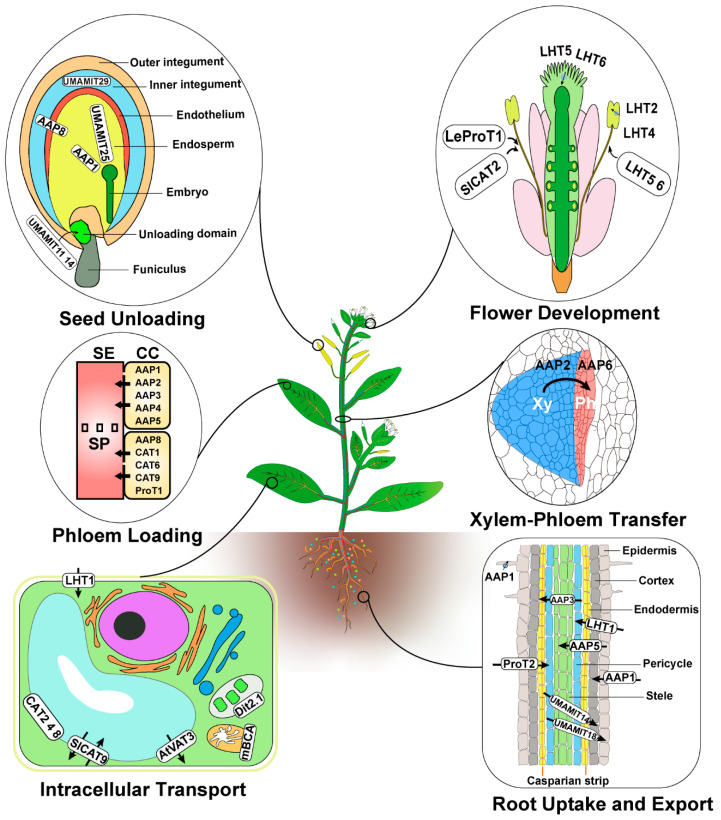
Summary of the role and site of action of characterized amino acid transporters in plants. The amino acid transporters mentioned in the main text play roles in root uptake and export, xylem–phloem transfer, intracellular transport, phloem loading, flower development, and seed unloading. Seed cross-section drawing is based on published micrographs [99,100]. Root section drawing is based on several reviews and research articles [1,12,48,104]. Black arrows cross rounded white boxes refer to the direction of transport when known. Information about the roles of each transporter highlighted here is provided in the reference list (Appendix A). Xy, xylem; Ph, phloem; SE, sieve tube; SP, sieve plate; CC, companion cell.

**Table 1 plants-09-00972-t001:** Overview of amino acid exporters in Arabidopsis.

Locus	Gene Name	Substrate (Expression in Yeast or/and *Xenopus* Oocytes)	Tissue Localization	Subcellular Localization	Phenotype	References
At2g40900	*UMAMIT11*	Glu and Gln	roots, leaves, flowers, throughout seed development	plasma membrane	*umamit11*: Levels of free amino acids in fruits (Asp, Thr, Glu, Ala)↑; Seed volume↓	[100]
At2g39510	*UMAMIT14*	broad range of amino acids	roots, leaves, flowers, throughout seed development	plasma membrane	*umamit14*: Levels of free amino acids in fruits (Ser, Asp, Thr, Gln, Glu, Asn, Pro, Ala, GABA)↑; Seed volume;↓ amino acid transferred to the roots and secreted by the roots↓	[8,100]
Atlg44800	*UMAMIT18 (SIAR1)*	Gln, Asp, Ala, Asn, Thr, Val, His, Leu	pericycle, stamen, developing seeds, roots	plasma membrane	*siarl* mutants: Early stages of silique development (amino acid content↓; anthocyanins↑); amino acid transferred to the roots and secreted by the roots↓	[8,17]
At1g25270	*UMAMIT24*	broad range of amino acids	developing seeds (seed coat)	tonoplast	*umamit24* knockout: Amino acid contents in seeds↓	[101]
At1g09380	*UMAMIT25*	broad range of amino acids	developing seeds (endosperm cells)	plasma membrane	*umamit25* knockout: Amino acid contents in seeds↓	[101]
At1g01070	*UMAMIT28*	Glu and Gln	roots, leaves, flowers, later in seed development	plasma membrane	*umamit28*: Levels of free amino acids in fruits (Asn, Pro, Ala, GABA)↑; Seed volume↓	[100]
At4g01430	*UMAMIT29*	Glu and Gln	roots, leaves, flowers, before late torpedo stage	plasma membrane	*umamit29*: Levels of free amino acids in fruits (Ser, Thr, Gln, Glu, Asn, Pro, Ala, GABA)↑; Seed volume↓	[100]
At1g75500	*UMAMIT5 (WAT1)*	IAA	developing xylem vessels and fibers	tonoplast	*wat1* mutants: A defect in cell elongation; no secondary cell walls in fibers	[18,109]
At2g01170	*BAT1*	Ala, Arg, Glu, Lys	vascular tissues	plasma membrane	*—*	[39,110]
At3g11900	*ANT1*	aromatic and neutral amino acid, Arg	flowers and cauline leaves	ER in the perinuclear region	*ant1* mutants: Essential amino acids within the SEs↑	[78,88,111]
At5g65990	*AVT3A*	neutral amino acids	various tissues of whole plants	tonoplast	—	[78]

At, *Arabidopsis thaliana*; UMAMIT, usually multiple acids move in and out; BAT, bidirectional amino acid transporter; ANT, aromatic and neutral transporter; AVT, amino acid vacuolar transport; Glu, glutamate; Gln, glutamine; Asp, aspartic acid; Ala, alanine; Asn, asparagine; Thr, threonine; Val, valine; His, histidine; Leu, leucine; Ala, alanine; Arg, arginine; Lys, lysine; Ser, serine; Pro, proline; GABA, γ-aminobutyric acid; IAA, indole-3-acetic acid; ER, endoplasmic reticulum; SE, sieve tube.

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
