# Peer review of "Amino Acid Transporters in Plants: Identification and Function"

_plants, 2020, doi:10.3390/plants9080972_

Round 1

Reviewer 1 Report

This review provides a great overview on the involvement / function of a large number of amino acid transports within the plants. The review is really worth to be published. I like the way by which the transport of amino acids has been described in detail. This review helps to understand the transport way and path of amino acids within plants very good.

I have a few comments, which may be considered prior publication:

  1. Line 25: ‘Following their uptake, nitrate and ammonium are reduced to amino acids in the root or in photosynthetically ………’. It is not correct that ammonium is reduced in ….. Within ammonium the N is reduced. Please rephrase.
  2. Line 39 to 48: May be a graph /picture explaining all the amino acid families and by this provides an overview, may be helpful for the reader.
  3. Line 63: ‘Most of the transporters characterized thus far localize to the plasma membrane and function as proton-coupled importers or exporters between cells’. Please explain the phrase …. exporters between cells …. a bit more. At this place it may be misunderstanding.
  4. In the next sentence (line 64) it is written ….. intracellular transport. Even this needs a bit more explanation.
  5. Line 126: ‘Ultimately, amino acids are released into the phloem apoplasm’. This is wrong. The phloem is not an apoplastic space. Sieve elements are controlled by CC cells and contained a so called mictoplasma; i.e. the vacuole is dissolved and combined with the cytosol. Please correct.
  6. Line 159: ‘To be exported out of leaves, amino acids are loaded into the minor veins of the phloem’. The term …‘minor vines of the phloem’… is misunderstanding. Veins always means xylem and phloem together. Do you mean minor vines of the leaves?
  7. Line 187; Beginning of the phloem unloading chapter: I think that here the way / process for the apoplastic phloem unloading needs a better description / more details. The release of amino acids into the apoplastic space prior they are taken up into sink cells is also be possible - according the apoplastic phloem unloading of sucrose. A generally symplastic unloading is not correct.
  8. Line 203: Here the meaning as well as the transport processes in the unloading domain is not clear enough. How can amino acids be accumulated? How should the transport from cell to cell be?
  9. Sentence line 207: ‘An analysis of umamiT11 and 206 umamiT14 mutants provided support for the physiological relevance of apoplastic transport, as these mutants accumulated free amino acids in fruits and produced smaller seeds’. How can the apoplastic transport looks like? Explanation?
  10. Line 219: ‘…..and might play an important role in providing amino acids during seed development’. How and which direction of seed development?
  11. Line 240: ‘…release into the leaf apoplasm…’ for what? Which process did you mean?
  12. Line 239 to 253: Here several UMAMIT transporters are explained. Because this transporter group seems important for the export of amino acids: Could a picture providing an overview be helpful?
  13. Line 261 GDU1: Is GDU1 really a transporter or a protein with regulatory function(s)?
  14. Line 292: To my opinion uptake of amino acids for the plants survive is not necessary. Root uptake of ammonium and nitrate may be sufficient for growth. This statement needs concretisation.

Reviewer 2 Report

This manuscript is a nice review of relevant and current literature pertaining to amino acid transporters in plants. The authors very clearly and concisely summarize the literature and highlight aspects of the field that require further study. I have just three main suggestions to improve the current manuscript before publication.

  1. While the authors do discuss some regulation in the manuscript, the major message is that not much is known about the regulation of these transporters. I would, therefore, recommend removing the word 'Regulation' from the title as it is slightly misleading. 
  2. There seems to be very little discussion of sulfur-containing amino acid transporters (Cys or Met) despite some evidence that Cys is transported between roots and shoots in some species (See https://link.springer.com/article/10.1023/A:1006148815106). I suspect some amino acids are transported as di- or tri-peptides (like gamma-EC or GSH). I recommend adding some specific discussion of the sulfur amino acids and how they might be transported if not via specific amino acid transporters. 
  3.  Section 6 (Transport Function In Carbon Metabolism) is a particularly interesting section as it introduces the idea that amino acid transport might impact NUE and photosynthesis. While most of our molecular knowledge of amino acid transporters comes from Arabidopsis, it is clear that C4 plants (which typically have a higher NUE than C3 plants) rely heavily on metabolite transport between cells and compartments to optimize their metabolism. I would recommend adding some discussion to this paragraph highlighting the need to identify and study amino acid transporters in the context of C4 organisms as this will more directly impact human nutrition and food security (many staple crops are C4). 
